# Metabolic Profile Characterization of Different Thyroid Nodules Using FTIR Spectroscopy: A Review

**DOI:** 10.3390/metabo12010053

**Published:** 2022-01-08

**Authors:** Vanessa Neto, Sara Esteves-Ferreira, Isabel Inácio, Márcia Alves, Rosa Dantas, Idália Almeida, Joana Guimarães, Teresa Azevedo, Alexandra Nunes

**Affiliations:** 1Department of Medical Sciences, iBiMED—Institute of Biomedicine, University of Aveiro, 3810-193 Aveiro, Portugal; vanessaneto98@ua.pt (V.N.); idalia24@ua.pt (I.A.); 2Centro Hospitalar do Baixo Vouga, CHBV—Endocrinology Department, 3810-164 Aveiro, Portugal; saragabrielaferreira@gmail.com (S.E.-F.); isabelmrinacio@gmail.com (I.I.); Marcia.Ines.alves@gmail.com (M.A.); rdantas84@gmail.com (R.D.); Joanaguimaraes.endoc@gmail.com (J.G.); tcmfazevedo@gmail.com (T.A.)

**Keywords:** nodular thyroid pathology, thyroid cancer, thyroid tissue, thyroid cytology, metabolome, Fourier-transform infrared spectroscopy, multivariate analysis

## Abstract

Thyroid cancer’s incidence has increased in the last decades, and its diagnosis can be a challenge. Further and complementary testing based in biochemical alterations may be important to correctly identify thyroid cancer and prevent unnecessary surgery. Fourier-transform infrared (FTIR) spectroscopy is a metabolomic technique that has already shown promising results in cancer metabolome analysis of neoplastic thyroid tissue, in the identification and classification of prostate tumor tissues and of breast carcinoma, among others. This work aims to gather and discuss published information on the ability of FTIR spectroscopy to be used in metabolomic studies of the thyroid, including discriminating between benign and malignant thyroid samples and grading and classifying different types of thyroid tumors.

## 1. Introduction

Thyroid cancer’s incidence has spiked in the last decades, although still maintaining an overall good prognosis [1]. However, its diagnosis can pose a challenge since it is difficult to differentiate a malignant nodule from a benign one [2,3]. Cytological analysis of fine needle aspiration (FNA) samples provides an estimated risk of malignancy, but definite diagnosis requires histological analysis, which has to be obtained through surgical resection. Thyroid surgery carries the risk for uncommon but significant complications, such as hypoparathyroidism or vocal cord paralysis. Only 20% of cytologically indeterminate nodules prove to be malignant, meaning a proportion of patients are subject to diagnostic rather than curative thyroidectomy. Therefore, further and complementary testing based in biochemical alterations may be important to correctly identify thyroid cancer and prevent unnecessary surgery. Since biochemical alterations precede morphological changes in cells, the cancer metabolome has gained relevance and may contribute to understand tumor biology and to identify early diagnostic biomarkers. Fourier-transform infrared (FTIR) spectroscopy is a metabolomic technique that, unlike histopathologic approaches, is rapid, low cost and uses a small amount of sample. This technique gives information on the biochemical composition of samples and allows the discrimination of samples with different metabolic profiles, being able to discriminate cancerous and non-cancerous samples. Spectral analysis is possible with multivariate analysis (MVA), which allows to identify groups of molecules that may act as potential biomarkers of cancer. FTIR has already shown promising results in cancer metabolome analysis of tissues, in the identification and classification of prostate tumor tissues and of breast carcinoma, and was also used in metabolomic studies of thyroid, enabling the discrimination between benign and malignant thyroid samples and grading and classifying different types of thyroid tumors. This works aims to gather the published data on the application of FTIR in the metabolome of the thyroid gland.

## 2. Thyroid Gland

The thyroid is an endocrine gland located in the lower anterior neck. Its main function is to secrete triiodothyronine (T3) and thyroxine (T4), hormones produced from the iodine obtained from a diet [4]. Thyroid function is regulated by the hypothalamic–pituitary–thyroid (HPT) axis. The hypothalamus produces thyrotropin-releasing hormone (TRH), which stimulates the pituitary gland to secrete thyroid-stimulating hormone (TSH), or thyrotropin. This hormone acts on the thyroid, stimulating cell growth and hormone production. Thyroid hormones, in turn, exert negative feedback on the hypothalamus and pituitary, inhibiting the secretion of TRH and TSH, in such a manner that the hormone levels are maintained within a narrow range [5].

## 3. Nodular Thyroid Pathology

Thyroid disorders are extremely common, making thyroid diseases one of the most common subtypes of endocrine disorders worldwide [6,7]. They can be divided into two independent entities: functional and structural disorders. Thyroid structural pathology is comprised mostly of nodular disease.

### 3.1. Epidemiology and Etiology

Thyroid nodules consist of lesions within the thyroid gland that are radiologically distinct from the surrounding thyroid parenchyma. These represent a common clinical entity, with up to 68% of the population having ultrasound-detectable nodules and higher rates being observed in women and the elderly [8]. In about half of the patients, the nodules are solitary [9]. Multinodularity (more than one node) increases with age, body mass index and female sex [10,11]. Even though they are prevalent, approximately 90% are asymptomatic [8]. The clinical relevance of thyroid nodules relates to the risk of thyroid dysfunction (5%), compressive symptoms (5%) or thyroid cancer (which occurs in about 10%) [12]. Some factors may contribute to a higher risk of malignancy, such as age, male sex, radiation exposure history in childhood or adolescence, family history of thyroid cancer or hereditary syndromes that include thyroid cancer (such as multiple endocrine neoplasia syndrome type 2 or familial adenomatous polyposis), rapid nodule growth or hoarseness [8,12]. Other risk factors were proposed, such as serum thyrotropin levels, thyroid antibodies, obesity and metabolic syndrome, but the evidence available at the moment is insufficient [13,14,15,16,17].

Functional thyroid disease occurs when there is either an under or overproduction of thyroid hormones, corresponding to a clinical state of hypothyroidism or hyperthyroidism, respectively [18]. This can be evaluated through measurement of serum T3, T4 and TSH. Thyroid nodules can occur with hypothyroidism, hyperthyroidism or euthyroidism. The function of the nodules can be independent from regulation from the HPT axis, meaning that the nodules can function differently from the remaining parenchyma. Hypofunctioning or “cold” nodules do not produce thyroid hormones, while hyperfunctioning or “hot” nodules synthesize hormones autonomously, without the need of stimulation from TSH, often resulting in hyperthyroidism.

### 3.2. Management and Prognosis

The main goal of thyroid nodules management is to identify the nodules at greater risk of malignancy, allowing for timely and effective treatment in order to prevent cancer-related morbidity and mortality, while preventing invasive procedures from being performed on benign or low risk pathology. More than 90% of thyroid nodules are benign and do not require immediate therapeutic intervention, only monitoring [9,19,20]. These nodules are rarely of clinical significance, except for very large nodules that result in compressive symptoms (from airway and/or digestive tract obstruction) or in hyperfunctioning nodules, which cause hyperthyroidism.

The vast majority of thyroid cancers (95%) correspond to differentiated thyroid cancer (DTC) that originated from thyroid follicular epithelial cells. The most common form of DTC, corresponding to 85% of patients, is papillary thyroid carcinoma, which carries the best overall prognosis. Follicular thyroid carcinoma accounts for 12% of DTC and poorly differentiated and anaplastic thyroid carcinomas, two entities associated with a worse prognosis, are much less frequent (<3% of DTC and <1% of thyroid cancers, respectively) [17,21,22].

Classically, surgical resection is the mainstay of thyroid cancer treatment—either lobectomy or total thyroidectomy, with or without lymph node dissection, according to preoperative staging [8]. Despite advances in surgical techniques aiming to minimize the occurrence of adverse effects, thyroid surgery still carries risks [23]. These include transient (1.2%) or permanent (0.6%) vocal cord palsy and transient (27.5%) or permanent (7.1%) hypoparathyroidism [24]. Complication rates are lower for surgeries performed by an experienced surgeon [23]. Besides these complications, permanent iatrogenic hypothyroidism, with need for lifelong supplementation with levothyroxine, occurs in all patients subject to total thyroidectomy and about 22% of patients following hemithyroidectomy [25].

Morbimortality associated with thyroid cancer derives mostly from complications of extrathyroidal extension, which occurs in 10–15% of patients, with invasion of adjacent structures or distant metastasis [17]. Locally advanced disease most commonly involves the strap muscles (53%), recurrent laryngeal nerve (47%), trachea (37%), esophagus (21%) and larynx (12%) [26]. The most frequent symptoms are dysphonia, dysphagia, dyspnea, coughing, bleeding, vocal cord paralysis, upper airway obstruction and cervical pain [27,28]. The most frequent sites of distant metastization are the lungs (75%), bone (45%) and brain (6.8%), with 25% of patients presenting with more than one affected organ [29]. However, it is still an entity with a predominantly good prognosis, with a 5-year survival rate of 98.3% overall and 99.9% in localized disease, making it the cancer with the highest relative survival rate in the USA between 2011 and 2017 [30].

### 3.3. Diagnosis

The more widespread use of imaging techniques and FNA biopsies resulted in a significant rise in the incidence of thyroid cancer, with a three-fold increase from 1975 to 2009. However, this has not resulted in an improvement in mortality rates, which remained relatively stable at 0.5 deaths per 100,000 [31]. The most likely explanation rests on the fact that most of the increment in incidence was a result of an increased diagnosis of small papillary carcinomas, which have been shown to be of very good prognosis and therefore are unlikely to lead to clinically apparent disease [32,33,34]. A study of thyroid cancer incidence trends estimated that over 830,000 women and 220,000 men might have been overdiagnosed between 2008 and 2012 in the 26 countries analyzed [35]. Besides direct complications from overtreatment, overdiagnosis leads to financial costs and an increased burden for health systems and for patients, with thyroid cancer survivors showing lower health-related quality of life and higher levels of anxiety, despite their good prognosis [35,36]. This knowledge has led to a recent trend towards a less invasive approach, in order to avoid complications of overdiagnosis and overtreatment of an otherwise indolent disease.

Since the majority of patients with thyroid nodular lesions are asymptomatic, the diagnosis is often accidental [27,28,37]. In the presence of a suspected or confirmed thyroid nodule, an ultrasound should be performed. Besides evaluating the presence of nodules, the sonographic features and size confer to a nodule a risk of malignancy, which is in turn used to guide FNA decision-making. Sonographic features associated with a higher risk of malignancy include the presence of microcalcifications, hypoechogenicity, irregular margins and a shape taller than wide measured on a transverse view [8]. Other ultrasound findings should be taken into account, such as the presence of suspicious lymphadenopathy, extrathyroidal extension, vascularity and a thick or absent halo [38]. Different recommendations have been issued by various scientific societies regarding indications for FNA [17,38,39]. Besides sonographic features, thyroid function should also be considered before FNA is performed. For patients with hyperthyroidism, a radionuclide thyroid scan, a functional study of the gland, should be obtained to assess whether the nodule is hyperfunctioning. Since these nodules are rarely malignant, cytologic evaluation can be dismissed [8].

The FNA specimens are then subject to cytopathological examination and its results are reported according to the Bethesda system. They are classified into one of six categories: I. Nondiagnostic or unsatisfactory; II. Benign; III. Atypia of undetermined significance or Follicular lesion of undetermined significance (FLUS); IV. Follicular neoplasm or Suspicious for a follicular neoplasm; V: Suspicious for malignancy; VI. Malignant. Again, an estimated risk of malignancy is attributed to each category, on which future treatment or follow-up decisions should be based. Definitive diagnosis of either benignity or malignancy can only be established through histological analysis, which requires that a surgical resection of the thyroid is performed. Nodules with a Bethesda II cytology have a low risk of malignancy, so no further immediate studies or treatment are required. Follow-up should be maintained according to sonographic features. For nodules within the Bethesda IV, V and VI categories, surgery is generally recommended, given the high risk of malignancy. For nodules with a non-diagnostic (Bethesda I) or FLUS (Bethesda III), repeat cytology should be considered. Nevertheless, in case of repeated results, in the presence of suspicious sonographic features or according to patient preference, surgery can be considered over ecographic monitoring [17]. However, only about 20% of indeterminate thyroid nodules (Bethesda III and IV) have proven to be malignant at final histology, meaning a significant proportion of patients are subject to a diagnostic rather than curative thyroidectomy [40]. With the intent of avoiding that phenomenon, multiple strategies have been proposed to enhance FNA cytology diagnostic performance, such as molecular testing [41].

## 4. New Diagnostic Approach—Metabolomics Techniques

Another promising approach includes “omics” techniques, which are used to accurately identify thyroid pathologies based on biochemical changes, with impact in the patients’ quality of life and in the efficiency of health services [42,43]. The “omics” field encompasses multiple approaches such as genomics, transcriptomics, proteomics, lipidomics and metabolomics. The latter is a vast, promising emerging field, since it allows the study of the complete set of biomolecules/metabolites that make up a biological sample, whether it is under physiologically or biochemically altered conditions due to the occurrence of diseases, such as cancer, or physiological processes, such as aging [44,45].

This study of small molecules can be applied in biological studies that use either tissues (e.g., ex-vivo samples, as biopsies) or in peripheral fluids (e.g., plasma, serum, urine, saliva), and can also be used in other in vitro studies, using cell line models [46].

Cancer is a pathology that is differentiated, especially by the manifestation of several cellular and molecular modifications, modifications that result in an altered metabolism in comparison with normal cells [43]. In this way, cancer cells manifest a unique and distinct metabolic phenotype [44]. Since biochemical alterations precede morphological changes in cells, the cancer metabolome, which represents the most “downstream” level of molecular life of a cell, has gained relevance and may contribute to identify early diagnostic and prognostic biomarkers to be applied in clinical practice. In addition to allowing the discovery of novel markers, the study of the thyroid cancer metabolome also enables an understanding of tumor biology through the study of the molecular pathways responsible for the process of carcinogenesis. Thus, the metabolome reflects the molecular and phenotypic state of an organism, both physiologically and pathologically, so the determination of metabolites in altered cancer cells seems to be a promising approach for its study [43,44]. Although this metabolomics approach is still not used routinely in clinics and pre-clinics, it allows to identify small molecules and, consequently, the phenotypic characteristics of cancer cells, as well as presents a series of characteristics that make it an excellent option, such as a quick, specific, sensitive and reproducible diagnostic analysis [47].

There are many studies in which the successful application of genomic [48,49], proteomic [50,51] and transcriptomic [49,52] techniques is evidenced in the area of thyroid cancer; however, the same situation does not apply to the application of metabolomics. Nevertheless, over the past few years, the metabolomics approach in terms of thyroid diagnosis has shown an enormous evolution, which is due, in part, to the application of several analytical techniques that allow the study of the complete set of small biomolecules that make up a biological sample. The main metabolomics techniques that have mostly been adopted in the thyroid, in an attempt to identify biomarkers, classify the thyroid cancer, facilitate an earlier diagnosis and prevent the development and invasion at the systemic level of the cancer itself, are Mass Spectrometry (MS) [53,54,55], Nuclear Magnetic Resonance (NMR) [53,56,57], Raman Spectroscopy [58,59,60] and Fourier Transform Infrared (FTIR) Spectroscopy [61,62,63,64,65,66,67,68,69,70,71,72,73,74,75,76,77,78,79]. This increase in the number of studies is attributed to the greater availability of improved and accurate metabolomics techniques that exist and also due to the development of appropriate statistical tools that are able to handle the huge amount of data resulting from the samples under analysis [80].

All of these metabolomics techniques, in general, present high sensitivity, specificity and precision with respect to the identification and often to the absolute determination of the concentration of the multiple metabolites that constitute the complex biological samples under analysis. In addition, these techniques also stand out for the fact that they are potentially non-invasive (in a biological fluid-based experimental design), have a high-throughput, are highly reproducible, non-destructive (some experiments use the sample without pre-processing, and thus do not cause any alteration/destruction of the sample that remains intact) and also allow the analysis of both solid samples (e.g., tissues) as well as liquids (e.g., cytologies or plasma) [43,53,81]. Thus, when these techniques are applied and analyzed correctly, clinically it is possible to obtain a reliable and an accurate result in relation to the identified thyroid pathology. However, there is an enormous complexity associated with the physiology and pathophysiology of the thyroid, as well as a great complexity and uncertainty, sometimes felt in relation to the metabolomics methods of diagnosis. This can result in the inappropriate use of these methods, culminating in an expensive (use of very expensive technical equipment) and, sometimes, time-consuming (slower sample processing) diagnosis [43,82].

### FTIR Spectroscopy and Thyroid Cancer

FTIR Spectroscopy is a metabolomic technique that can be used as a diagnostic method in the area of cancer. In addition to what was mentioned above about the other metabolomic techniques, FTIR spectroscopy is still characterized as being reagent-free (does not require any biochemical reagents), quite simple (can be performed by any clinician or technician with only a minimal training), fast (it only takes 15 min to perform) and objective (allows a reduction in the mistakes made by human subjective judgment), which requires little or no sample preparation, requires a small amount of sample (leaves sufficient material for other clinical tests) and that has proven to be a cost-effective technique whose application can be extended to clinical diagnosis [62,67,68]. It is also a very innovative method that allows the screening of the metabolic profile of each analyzed sample and, consequently, to trace its spectroscopic profile. The IR spectrum, obtained through FTIR analysis, can serve as a spectroscopic signature that allows the characterization of several pathologies depending on their metabolic profile [47]. Despite this huge set of advantages, FTIR presents a major disadvantage or limitation that is based on the intense absorption of water in the mid-IR region of the electromagnetic spectrum, masking the vibrational absorption of the other biomolecules, this is a limiting situation due to the fact that all biological samples (tissues, cells and body fluids) that can be analyzed are mostly made up of water [83]. However, this situation can be easily overcome by three different strategies, namely, by subtracting the water absorption signal, by dehydrating the sample or letting the sample dry directly on the crystal when attenuated total reflectance (ATR) is used as a sampling method [47,84].

Some research groups have already shown promising results in which the high sensitivity of this spectroscopic method was detected in the identification and discrimination of various types of cancer at the level of the spectroscopic analysis of tissue samples from organs such as the lungs [85], breast [86] and prostate [87,88]. In fluid analysis, FTIR has been used in the study of the metabolome associated with physiological alterations, revealing a huge potential in the early and minimally invasive diagnosis of several pathologies [89,90]. In fact, FTIR spectroscopy is a promising metabolomic analytical technique; it is a type of vibrational spectroscopy that is based on the vibration of the atoms in a molecule [47]. This technique makes it possible to detect the biochemical composition of a biological sample in relation to the different macromolecules that constitute it, such as nucleic acids, carbohydrates, lipids and proteins, by enabling the identification of functional groups, molecular conformations, types of connection and the different intermolecular interactions that make up the sample. It is then possible to use this technique to study the composition of more complex biological samples such as cells, tissues and body fluids [91]. Each chemical or biochemical sample has its own infrared spectrum which is reflected in the existence of a unique infrared fingerprint attributed to each analyzed substance. Thus, each change that occurs in the biomolecules, resulting from the process of carcinogenesis, for example, results in the modification of this “fingerprint” and, consequently, allows the distinction between normal cells and altered cancer cells [47,91].

According to the inclusion criteria defined for the construction of Table 1, less than 20 papers were found in the last 20 years. These studies include both thyroid tissue and cytology samples. The small number of existing papers may be related to the fact that cytological samples are difficult to collect, due to the invasive nature of the FNA procedure, and to the small amount of sample obtained within this process. However, though more challenging, the use of cytology samples, in our opinion, is more advantageous, since their analysis, realized prior to surgery, may contribute to avoid thyroid excision in patients with an unprecise diagnosis.

Four of the studies from Table 1 used microspectroscopy, a technology not available in all metabolomics laboratories. This approach could be advantageous for the analysis of anatomical pieces, to identify normal and tumor profiles and also to detect the characteristic profiles of the transition region around the tumor. However, studies based on ATR or transmission are quite promising. They require small amounts of sample and are able to accurately discriminate normal from tumor samples.

Regardless the analysis tool and the type of sample used, these metabolomic studies can be an advantage in the quality of life of patients. An accurate analysis of a cytological sample can provide a precise classification result that avoids an unnecessary thyroidectomy. Additionally, metabolic changes precede cytological changes, which allows early identification of a tumor in cytological samples.

In our opinion, before spectroscopy being introduced into clinical practice, and to be used in the diagnosis and classification of thyroid specimens, there is still plenty of work ahead. The dimension of the data sets used needs to be increased. Only four of the studies identified in Table 1 have a number of participants higher than 100. It is essential to have large data sets to be able to draw conclusions about the spectroscopic/metabolic profiles of the samples.

In recent years, the number of spectroscopic studies that uses multivariate analysis has been increasing. These tools are essential to analyze larger data sets and to extract relevant and systematic information from spectroscopic data. Only with these statistical tools it will be possible to create classification algorithms that will accurately enable the use of spectroscopy in clinical practice. It is also crucial to use validation data sets to confirm the information obtained by the classification algorithms. Studies that have developed classification algorithms and that use independent data sets to validate the results are uncommon. Therefore, there are still some points to explore in the metabolomic study of thyroid nodules.

The mid-IR region can be divided into four distinct spectra regions, namely, the X-H stretching region (4000−2500 cm^−1^), the triple-bond region (2500−2000 cm^−1^), the carbonyl (around 1700 cm^−1^) and protein region (1700−1500 cm^−1^) and the fingerprint region (1500−600 cm^−1^) [92].

Associated with these spectra regions, important with regard to the analysis of biological samples, are the main biomolecules that make up these samples: lipids (3000−2800 cm^−1^); proteins (1800−1300 cm^−1^), which can further differentiate into two specific bands, namely, amide I (1700−1600 cm^−1^), the most used in these analyzes, and amide II (~1540 cm^−1^), both quite sensitive to the secondary structure of proteins; phospholipids (~1740 cm^−1^); amino acid side chains and fatty acids (1480−1300 cm^−1^); and carbohydrates and phosphates associated with nucleic acids (1300−900 cm^−1^) [93]. Any change that occurs in the frequency (increase or decrease) or in these spectroscopic bands (shape or intensity) may be indicative of cellular changes in the analyzed sample [84,93].

Taking into account all the information mentioned above, a typical IR spectrum of a normal thyroid cytological sample, obtained over the wavenumber 4000−600 cm^−1^, is presented in Figure 1. The spectral peaks of the typical thyroid cytology and the corresponding biochemical components and vibrational modes are shown in Table 2.

FTIR spectroscopy is widely used nowadays in the most diverse scientific areas and much emphasis has been given to the application of this technique as a possible method of early diagnosis of cancer. Many promising results have been demonstrated, since this spectroscopic technique appears to be a better approach compared to traditional diagnostic methods (histochemical methods), which are more expensive and more time consuming. Although, by itself it constitutes a technique with a great potential for diagnosis, in order to extract biological information from the data, it is necessary to have a greater knowledge of the spectroscopic characteristics of the different nodular lesions, both benign and malignant, and, simultaneously, the development of complementary statistical methods that allow to differentiate with greater sensitivity and specificity the different samples analyzed corresponding to these pathologies [62,67,68,72]. It is unquestionable that the quality and relevance of the information collected through spectral analysis and the potential application of the information collected in clinical practice, such as diagnosis and prognosis, is only possible using multivariate analysis tools. Additional information about these tools and their application in data analysis can be found in [94,95,96].

As previously mentioned, in order to improve the prognosis of patients who present thyroid carcinomas, and taking into account that the majority of these nodular thyroid lesions are receptive to medical and surgical management, it is important to use diagnostic methods that allow their early detection; FTIR spectroscopy seems to be an excellent option to be applied in this context [47]. The available thyroid cancer studies discriminate and characterize both healthy and pathological thyroid samples and the analysis is performed in tissue anatomic pieces obtained after thyroidectomy [61,66,67,68,69,70,72,73,74,75,76,77,78,79], in cytological samples obtained from the realization of FNA biopsies [62,72,75] or even through the application of the FTIR spectroscopy technique in a non-invasive way; that is, applied to the skin surface on thyroid nodules [63,64,65,71].

According to Table 1, one of the first studies carried out involving the FTIR spectroscopy technique in the diagnosis of thyroid neoplasms showed the presence of thyroglobulin, collagen fibers and some bands related to unknown proteins in the analyzed samples. This study allowed to conclude that although a minor overlap between the bivariate analysis histograms of neoplastic tissues and normal tissues exists, this technique can efficiently allow the distinction between cancer cells and normal cells of the thyroid gland [75]. Integrating the information from Table 1 and Table 2, this distinction and therefore the confirmation of the malignancy, of certain thyroid samples was also observed in other studies, in which the FTIR spectra of nodular goiter (NG) tissue and the papillary thyroid cancer (PTC) were compared, and significant differences in proteins (1700−1500 cm^−1^), lipids (4000−2500 cm^−1^ and 1700 cm^−1^) and nucleic acids (1200−600 cm^−1^) were found [63,68]. However, there is still a study that, despite showing significant differences between the spectra of the NG and thyroid carcinoma groups, similarly to previous studies, in practically all regions of the spectrum demonstrates the inexistence of these differences in the protein region (amide I and amide II signals) [77]. In this process of differentiation between normal, benign and/or malignant thyroid samples, there are studies that show the great importance of DNA and of some small metabolic products, as DNA metabolism is increased in tumor samples, as verified, although sometimes with less significance, with proteins [62,67,69,79]. Accordingly, there are several studies that also show an alteration in lipid metabolism in tumor thyroid tissues, due to the increased consumption of lipids (differences observed in 4000−2500 cm^−1^ and 1700 cm^−1^), as well as carbohydrates (differences observed in fingerprint region: 1500−600 cm^−1^), by these malignant tissues, thus resulting in a decreased absorbance when comparing malignant thyroid samples with the normal ones in the regions corresponding to these biomolecules [63,67,70,79]. Another aspect presented in different studies that allow the distinction between cancerous and normal thyroid samples is related to the protein secondary structure evidenced. The existence of a β-sheet structure in cancerous samples has been demonstrated, while normal samples show an α-helix structure, these structural differences can be evaluated thought the location of the spectroscopic signal around 1600 cm^−1^ [61,79]. Although FTIR can be used to analyze plasma for metabolic and spectroscopic profiling there are no studies using the plasma of nodular thyroid pathologies patients.

## 5. Conclusions

Thyroid nodular disease is one of the most recurrent endocrine disorders worldwide, and in recent decades its incidence and prevalence have increased considerably. Although only a minority of these lesions present a malignant character, it is crucial to carry out an early diagnosis. One of the very simple and fast diagnostic techniques, even considered the gold standard technique in this process, is the FNA biopsy technique. Despite its routine clinical applicability, this technique is quite often associated with diagnostic errors that result from inconclusive, indeterminate and false-positive cytological results. To counteract the high percentage of thyroids removed unnecessarily due to misdiagnosis, the application of metabolomic techniques in the diagnosis of thyroid nodules seems to be a promising alternative. By allowing the study of all biomolecules that make up the sample under analysis, any metabolomic alteration that may occur is likely to be detected, which could result in the consequent identification of possible early biomarkers for the diagnosis of thyroid cancer. The FTIR spectroscopy technique is one of the metabolomic methods that has shown to be very promising with regard to the diagnosis of various types of cancer, including of the thyroid gland. Characteristics such as its simplicity, speed, objectivity, non-invasiveness, lack of use of reagents, little need for preparation and the small amount of sample needed make this technique an excellent option to be routinely applied in clinical diagnosis in the future. Since this innovative technique allows the screening of the metabolic profile of the analyzed samples and, consequently, allows to trace the spectroscopic profile of each of these samples, it may be possible, through its application, to categorize a vast number of analyzed samples in different types of thyroid nodules according to their spectroscopic signatures. In order for this methodology to be tested and validated for its effectiveness, it is necessary to create different cohorts representing the different types of thyroid nodules under analysis. Although it is essential to carry out further studies for its clinical application, FTIR spectroscopy promises to be a very helpful method in the area of thyroid cancer diagnosis.

## Figures and Tables

**Figure 1 metabolites-12-00053-f001:**
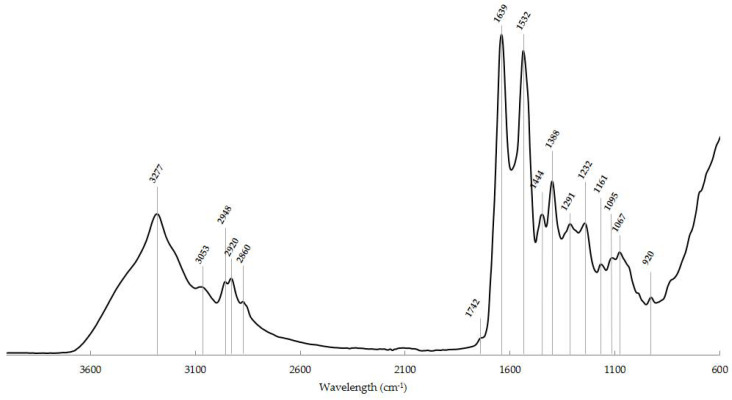
Typical IR spectrum of a normal thyroid cytological sample in the ~4000−600 cm^−1^ region with the main spectroscopic signals. *x*-axis: wavenumber (cm^−1^); *y*-axis: arbitrary units (a.u.).

**Table 1 metabolites-12-00053-t001:** Summary of the FTIR spectroscopy research studies in the characterization of normal and cancerous thyroid samples, highlighting the sample and analysis types, acquisition parameters and the principal findings of each study. Search for studies performed on 22 December 2021, on PubMed, under the following terms: (“Thyroid Gland”[MAJR]) AND (“Spectroscopy, Fourier Transform Infrared”[MeSH]); and on ScienceDirect, under the following terms: (Infrared Spectroscopy AND Thyroid tumor AND FNA cytology). The results were manually filtered to ensure that they all reported studies that use FTIR spectroscopy in the analysis of normal and nodular thyroid samples. Articles in Chinese and academic research documents were excluded from the table. The sequence of the articles in the table follows a chronological order. (↑ increase; ↓ decrease).

Sample Type	Acquisition Conditions	Analysis Type	Main Findings	References
FNA citology (original aspirate; cell-free supernant; cell pellet)	µ-FTIR Scans: 256 Resolution: 4 cm ^−1^ Spectral region: 1200−900 cm^−1^	Unsupervised Cluster analysis (Ward’s minimum variance algorithm and Euclidian distances) and Bivariate statistical analysis;	- NG spectra: strong protein (1545 and 1655 cm^−1^) and carboxylate bands (1409 and 1578 cm^−1^); Typical follicle characteristic spectra: colloid: α-helical protein profile of thyroglobulin; epithelium: DNA (968 cm^−1^), collagen (1338 cm^−1^) and lipids (1740 cm^−1^); - Neoplastic tissues: ↑DNA ↓Protein (except: PTC that ↑Protein);	[75]
Tissue from thyroid gland	µ-FTIR Scans: 16			
*n* = 89 FNA citology samples of 89 thyroid nodules (original aspirate; cell pellet) Thyroid powder	Transmission Resolution: 4 cm^−1^ Spectral region: 1800−900 cm^−1^	Baseline correction, area normalization and second derivatives using the Savitzky-Golay algorithm; Unsupervised Cluster analysis (Ward’s minimum variance algorithm and Euclidian distances) and Supervised Linear discriminant analysis;	- The most prominent bands are protein absorptions (1652 and 1542 cm^−1^); - Thyroid powder: high DNA content (966, 1087, 1240 and 1713 cm^−1^); - Cell pellets - more content of lipids (1740 cm^−1^); - Thyroglobulin and FNA aspirate - characteristic carbohydrate bands (1000–1190 cm^−1^) and also two strong protein bands (1545 and 1655 cm^−1^); - Tumor group - prominent carboxylate (1409 and 1580 cm^−1^), lipid ester (1740 cm^−1^) and carbohydrate bands (950–1200 cm^−1^);	[62]
*n* = 184 Cervical lymph nodes (61 metastatic; 123 non-metastatic) from 22 PTC patients	ATR Scans: 32 Resolution: 8 cm^−1^ Spectral region: 4000−1000 cm^−1^	Intensity measurement of spectroscopic signals with relative intensity ratios; Wilks’ lambda linear discriminant analysis;	- Metastatic group: ↑ protein (3280, 1640 and 1546 cm^−1^) and DNA/RNA (1240 cm^−1^); ↓ lipids (2925, 2855 and 1743 cm^−1^) and carbohydrates (1165 cm^−1^); - Peak position differ between metastatic and non-metastatic lymph nodes;	[67]
*n* = 60 60 Thyroid gland tissue samples (43 of NG; 17 of PTC)	ATR Scans: 32 Resolution: 4 cm^−1^ Spectral region: 4000−800 cm^−1^	Baseline correction, straight-line generated and smoothed; Measured the wave intensity ratios and the peak positions; canonical discriminant analysis	- Differences between the spectroscopic profile of benign and malignant groups; - Main differences between PTC and NG groups: peak positions: P1640 (amide I), P1240 (P = O stretch), P1550 (amide II); peak intensity ratios: 3375/1460, I1640/1460, 1400/1460, 1550/1080, 1080/I1460, and 1640/1550;	[68]
*n* = 80 80 Thyroid gland tissue samples T3 and T4 Thyroid hormones	µ-FTIR Resolution: 4 cm^−1^ Spectral region: 4000−750 cm^−1^	Vector normalization, converted to second derivatives using the Savitzky–Golay algorithm, noise-filtered and phase corrected; Hierarchical cluster analysis (Ward’s minimum variance algorithm and Euclidian distances);	- T3 spectra: peaks at 916, 1630 and 1460 cm^−1^; - T4 spectra: peaks at 1190, 1476 and 1468 cm^−1^; - Diodotyrosine spectra: strong peak at 1468 cm^−1^; - Peak at 1468 cm^−1^ (amount of DIT) and at 1460 cm^−1^ (amount of MIT) found in all spectrums of normal thyroid tissues but showed differences in their intensity profiles; - Iodinated thyroglobulin: α-helix structure; - Non-iodinated thyroglobulin: β-sheet structure;	[73]
*n* = 161 161 subjects (25 males; 136 females)—111 patients undergoing thyroid surgery; 50 healthy volunteers	ATR using optical fiber trough skin surface Scans: 32 Resolution: 8 cm^−1^ Spectral region: 3100−1000 cm^−1^	Baseline correction and a smoothing with a 9-point moving average; Intensity measurement of spectroscopic signals with relative intensity ratios; Wilks’ lambda stepwise discriminant analysis	- Normal control group successfully dicriminated from NG and PTC groups (High discriminant accuracy (88,8%); - Potential to noninvasively discriminate thyroid nodules (NGs and PTCs) from normal controls; - Difficult to discriminate NG from PTC group; - In cancer groups: ↓ relative intensity ratios for carbohydrates and lipids;	[63]
*n* = 20 20 Thyroid gland tissue samples (10 of FvPTC; 10 of FTC)	µ-FTIR Scans: 8 Resolution: 4 cm^−1^ Spectral region: 3850−900 cm^−1^	Baseline correction; Normalization of average spectra to the absorbance of 1651 cm^−1^; Principle component analysis and Linear discriminant analysis;	- Spectral analysis of follicular cells discrimination between FvPTC and FTC; - It wasn’t possible to discriminate FvPTC from FTC;	[76]
*n* = 112 112 Thyroid gland tissue samples (67 of NG; 10 of thyroid carcinomas)	ATR Scans: 32 Resolution: 8 cm^−1^ Spectral region: 1900−1050 cm^−1^	Intensity measurement of spectroscopic signals with relative intensity ratios; Tests of normal distribution and variance of homogeneity and Student’s *t* test	- Significant differences between spectra of NG and thyroid carcinomas in almost all peaks (except protein peaks);	[77]
*n* = 44 44 Thyroid gland tissue samples (16 of NG; 3 of FA; 4 of thyroiditis; 19 of PTC; 2 of FTC)	Not indicated	Normalization of the spectra to the absorbance of 1651 cm^−1^; Second derivative; Determination of bands areas; Calculation of relative areas ratio; one-way ANOVA and the Tukey and Fisher tests	- Possible to discriminate between: benign nodules from healthy tissues (using spectral region 1345–1482 cm^−1^); malignant nodules from healthy tissues (using mean diameter of 1240 cm^−1^);	[78]
*n* = 15 Thyroid gland tissue samples of 15 WI-FTC patients	ATR Scans: 64 Resolution: 2 cm^−1^ Spectral region: 4000−400 cm^−1^	Baseline correction (concave rubberband method), normalization (min.-max. method) and second derivative; Measurement of area under the curve; Principle component analysis and Linear discriminant analysis;	- Main difference between normal and tumor tissue in the protein region (concentration and structure): β-sheet in cancerous; α-helix in normal tissues; - Malignant tissue: values of maximal absorbance are ↓ in all spectra regions; - Neoplastic samples: change in lipid metabolism;	[79]
*n* = 32 32 Thyroid gland tissue samples (15 FTC patients and 17 FA patients)	ATR Scans: 32 Resolution: 2 cm^−1^ Spectral region: 1800−800 cm^−1^	Baseline correction and normalization of average spectra to the intensity sum of all peaks; Principle component analysis and Linear discriminant analysis;	- Neoplastic tissue spectra - ↓ values of absorbance in protein, nucleic acids and lipids regions compared with normal thyroid tissues (↓ is less significant for FA than for WI-FTC); - ↑ collagen groups in neoplastic tissues; - Protein secondary structure: β-sheet in cancerous; α-helix in normal tissues;	[61]
*n* = 14 14 Thyroid gland tissue samples (3 of metastatic lymph nodes PTC (PTC+); 6 of non-metastatic lymph nodes PTC (PTC-); 5 of normal tissue)	ATR Scans: 48 Resolution: 4 cm^−1^ Spectral region: 4000−600 cm^−1^	Baseline correction, vector normalization, first deivative and smoothing; Principle component analysis and Linear discriminant analysis; Leave one out cross validation;	- Normal group successfully discriminated from PTC+ and PTC- by intensity differences of phosphate bands associated with nucleic acids, proteins and lipids; - PTC+ and PTC- primary tumors are differentiated by their nucleic acid, protein and lipid content; - Clear separation of normal, PTC+ and PTC- groups through Principle component analysis;	[69]
*n* = 164 164 Thyroid gland tissue samples (76 malignant; 88 benign)	ATR Scans: 64 Resolution: 8 cm^−1^ Spectral region: 4000−1000 cm^−1^	Normalization (z-score normalization) and baseline correction (rubberband method); Principle component analysis and Linear discriminant analysis; Shapiro–Wilk test, tests of variance of homogeneity and Mann–Whitney U test;	- Significant differences between spectra of benign and malignant tissues; - Malignant tissues: ↑ absorbances of peaks attributed to amides and phosphorylated proteins; - Benign tissues: ↑ absorbances of peaks related to lipids, nucleic acids, carbohydrates and glycogen; - Phosphorylated proteins (878 cm^−1^ and 880 cm^−1^) are the most dominant component (84,65% variability among the samples);	[70]

**Table 2 metabolites-12-00053-t002:** Spectral peaks and the corresponding biochemical components and vibrational modes of a typical normal thyroid cytological sample obtained by FNA. Adapted from [47,73,78,86,88,91].

Wavenumber (cm^−1^)	Assigned Biochemical Component	Vibrational Mode
3277	Amide A: peptide, protein	N-H stretching
3053	Amide B: peptide, protein	N-H stretching
2948	Lipids	CH_3_ asymmetric stretching
2920		CH_2_ asymmetric stretching
2860		CH_2_ symmetric stretching
1742	Phospholipid esters	C = O stretching
1639	Amide I: parallel β-sheets	C = O stretching, C-N stretching, in-plane N-H bending
1532	Amide II	N-H stretching, C-N stretching, C-C stretching
1444	Membrane lipids and proteins	CH_3_ and CH_2_ deformation
1388	Phospholipid, fatty acid, triglyceride	CH_3_ symmetric wagging
1291	Amide III	N-H bending, C-N stretching, C = O stretching, C-C stretching, CH_3_ stretching
1232	Nucleic acids	PO_2_—symmetric stretching
1161	Carbohydrates	C-O stretching
1093 and 1067	DNA, RNA, phospholipid, phosphorylated protein	PO_2_—symmetric stretching
920	Dianionic phosphate monoesters of phosphorylated proteins and nucleic acids	PO_3_ ^2−^ symmetric stretching

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
