# Peer review of "Metabolic Profile Characterization of Different Thyroid Nodules Using FTIR Spectroscopy: A Review"

_metabolites, 2022, doi:10.3390/metabo12010053_

Round 1
Reviewer 1 Report
The manuscript “Metabolic Profile Characterization of Different Thyroid Nodules Using FTIR Spectroscopy: a Review” focuses on the applications of FTIR spectroscopy as metabolomic tool for the study of normal and cancerous conditions in the thyroid. The idea behind the review is interesting as the specific applications of IR spectroscopy in this field are few (compared for example to the characterization of other pathological conditions such as colon, breast, lung cancer) while their possible outcome would be of great interest in limiting thyroid resection to cases that truly require it. The introduction is well written and clear. Summarizing the main points of the papers reviewed in a compact, easy-to-reference table is, in my opinion, an added value to the review. However, the main part of the work regarding FTIR applications appears to be not fully developed as the results obtained in the selected literature are discussed superficially. This same part is also poorly edited in my opinion.
An accurate revision of section 4 is therefore needed before I can recommend the work for publication. I suggest grouping all information about FTIR in subparagraph 4.1, moving content from lines 237-265 into it, and merging the general overview of the technique to avoid repetitions. This part should also be shortened to give more space to the discussion of FTIR results in thyroid cancer research, which should be further developed as it is the main part of the paper, according to the chosen title.
Subsequent issues should also be addressed:
- Table 1
- Caption: Why is the research only updated to March 2021? More important, which were criteria used for the selection of the relevant references? Quick research in Pubmed using the terms indicated by the authors revealed more apparently relevant papers with respect to those included in the review, could you please explain?
- Acquisition conditions: The FTIR mode used for the analysis would be interesting information to add to this section, especially when it differs from the classical transmission mode. For example, in ref. 105 transflection (transmission/reflection) mode is selected. Could you add this information where missing?
- Analysis Type: The third column reports the type of data analysis adopted in the paper, which I think is very interesting and useful for the readers. However, in the text, there are only generic references to multivariate analysis and consequently, the approaches described may be unclear. Could you suggest an appropriate reference (perhaps a review) that describes multivariate data analysis for FTIR applications in greater detail?
- Main Findings: The acronyms NG and PTC have never been made explicit in precedence, nor are they in the rest of the text.
- References: What is the criterion adopted for the organization of the references in the table? Why are they not listed in order of appearance, since all those concerning FTIR application to the study of thyroid nodules are introduced simultaneously on page 5 line 209? More importantly, why are the results of some of the cited papers (99,100,101,104,106) not included in the table? If they are not relevant, they should be removed from the bibliography; if they are, they should be included in the table and discussion.
Minor issues:
- 5, lines 207-209: There are too many references for metabolomics techniques that are not of primary interest for the review. I suggest limiting the bibliography for MS, NMR and Raman spectroscopy to a recent review, if any, and possibly one or two more significant papers.
- 5, line 214: change “non-destructible” with “non-destructive”. This is not completely true for MS, but since the quantity of sample required for the analyses is small, I guess the statement can be considered somehow acceptable.
- 5, lines 225-235: this paragraph is redundant as it is a repetition of what has been said a few lines above. The consideration of the development of appropriate statistical tools could be introduced together with the general presentation of the metabolomic techniques.
- Pag 9 line 289: the -1 of cm-1 must be apex. The same in the following lines (289-301) and in Fig 1 caption.
- 9 line 302: "Figure 2" should be "Figure 1".
- 9, line 303: “Table 7” should be “Table 2”
- Figure 1: “AU” text box covers part of the spectrum. The axis label is usually shown to the left of the spectrum and indicated as “Absorbance (a.u.)”. The decimal digits shown for the wavenumbers are poorly significant and make the figure confusing; they should also be a bit larger to be well readable.
- 10, lines 322-323: check the sentence “being whereas FTIR spectroscopy seems to be an excellent option to be applied in this context”
- 10, lines 323 and 330: “table 6” should be “table 1”
- 10, line 334: consider changing “histograms” in “bivariate analysis histograms”
- 11, line 339-342: the meaning of this sentence is not clear, please rephrase.
- References: check reference style to meet the journal requirements.
-
- It is not possible to identify ref. 1. Add the needed information.
- Volume and page range are missing in most of the references. DOI should also be added.
Reviewer 2 Report
The manuscript presents a review of the use of FTIR Spectroscopy for metabolic profile vharacterization of different thyroid nodules. This is an important emerging area of diagnostics, and the review is tomely and appropriate. Ingeneral the review is well strcutured and presented, although there is a number of issues whch should be considered to improve the quality of the presentation.
(i) although generally good, the manuscript would benefit from a thorough proof reading for language and grammar.
(ii) Abstract - "This work aims to..." and also in relation to the end of the manuscript, the wothors should do more than just gather information about what has been published, but should also give an informed, expert opinion about what is the curent state of the art (e.g. how good are curent FTIR techniques compared to establshed techniques?) and what should be done to improve or better establish FTIR techniques?
(iii) "Thyroid cancer’s incidence has spiked in the last decades, though maintaining an overall good prognosis. However, its diagnosis can pose a challenge." - please provide reference
(iv) 4. New diagnostic approach – Metabolomics Techniques - how are metabolomics studies currently carried out? As a reader, I wanted to know, as I was reading, whether these are commonly in vivo, ex vivo (biopsies, biofluids)?
(v) "non-invasive", "non-destructive" are terms often used in relation to biospectroscopy, but if they require the removal of tissue, they are at least invasive.
(vi) "There are many studies in which the successful application of genomic, proteomic and transcriptomic techniques is evidenced in the area of thyroid cancer... " please reference some of these.
(vii) "Despite this huge set of advantages, FTIR pre-250 sents a major disadvantage or limitation that is based on the intense absorption of water in the mid-IR region of electromagnetic spectrum, masking the vibrational absorption... " Please provide a reference for this. Water principally absorbs from ~1550-1650 cm-1 and 3000-3400 cm-1. It does not mask all molecular vibrations.
(viii) "Although few studies use FTIR to study thyroid cancer, as summarized in Table 1, .." The main body text should briefly summarise the information in the caption of Table 1, abnd state how many studies were found, and for example what type of samples they were measured on.
(ix) As stated in the following paragraph, the "double bond region (2000-1500 cm−1)" is critically important for protein analysis, and is not considered normally as "the double bond region".
(x) Absorbance is the log of a ratio, and so the Y-axis of Figure 1 should have no units, not arbitrary units.
Reviewer 3 Report
1. A very controversial issue of attributing Fourier-IR spectroscopy to metabolomic technologies. Nevertheless, the analysis by this method provides information on the qualitative composition; its application for diagnostics is possible only in conjunction with multivariate statistical analysis. 2. The authors give an example of a spectrum of normal thyroid tissue. Cancer tissue should also be brought in. What changes are observed in the transition zone around the tumor? Can IR spectroscopy analyze tumor boundaries? 3. There is no analysis and generalization of table 1. It is necessary to formulate what general changes are characteristic for thyroid cancer, for nodular goiter? I would advise to add the summarized data to table 2 with an additional column. 4. It is necessary to describe the statistical methods that I use to process the results of IR spectroscopy in thyroid cancer.
Round 2
Reviewer 1 Report
I appreciate the improvement work done by the authors on the text. The organization appears clearer and the data commentary more personal.
Reviewer 3 Report
The authors responded in detail to the comments of the reviewer and significantly revised the manuscript. In its present form, the article can be recommended for publication.